# Conformation-Specific Association of Prion Protein Amyloid Aggregates with Tau Protein Monomers

**DOI:** 10.3390/ijms24119277

**Published:** 2023-05-25

**Authors:** Mantas Ziaunys, Kamile Mikalauskaite, Lukas Krasauskas, Vytautas Smirnovas

**Affiliations:** Institute of Biotechnology, Life Sciences Center, Vilnius University, LT-10257 Vilnius, Lithuania

**Keywords:** amyloids, prion proteins, Tau protein, protein aggregation, fibril structure

## Abstract

Protein aggregation into amyloid fibrils is associated with several amyloidoses, including neurodegenerative Alzheimer’s and Parkinson’s diseases. Despite years of research and numerous studies, the process is still not fully understood, which significantly impedes the search for cures of amyloid-related disorders. Recently, there has been an increase in reports of amyloidogenic protein cross-interactions during the fibril formation process, which further complicates the already intricate process of amyloid aggregation. One of these reports displayed an interaction involving Tau and prion proteins, which prompted a need for further investigation into the matter. In this work, we generated five populations of conformationally distinct prion protein amyloid fibrils and examined their interaction with Tau proteins. We observed that there was a conformation-specific association between Tau monomers and prion protein fibrils, which increased the aggregate self-association and amyloidophilic dye binding capacity. We also determined that the interaction did not induce the formation of Tau protein amyloid aggregates, but rather caused their electrostatic adsorption to the prion protein fibril surface.

## 1. Introduction

Protein aggregation into insoluble, highly structured fibrils is associated with the onset and progression of multiple amyloidosis, including neurodegenerative Alzheimer’s, Parkinson’s or prion diseases [1,2]. The process of such fibril formation has been a subject of intense research for multiple years, which resulted in a general understanding of the fundamental aspects of protein/peptide assembly into amyloid structures [3]. However, the aggregation mechanism is still not fully understood and reports of potential new fibrillization steps or interactions appear every year [4,5,6]. This, in turn, has severely impeded the search for cures, with only a handful of effective treatments of particular amyloid-related diseases being currently available [7,8]. Considering the ever-increasing number of individuals affected by amyloid-related disorders, it is imperative to gain a better understanding of the protein aggregation process.

Over the last few years, there has been an increasing number of reports regarding the phenomenon of amyloidogenic protein cross-interactions during amyloid fibril formation. Some of the observed cross-interactions involved proteins/peptides related to widespread neurodegenerative disorders: amyloid-beta with alpha-synuclein [9], Tau [10,11] and prion protein [12], as well as alpha-synuclein with prion protein [13] and Tau [14,15]. Other amyloidogenic proteins, such as lysozyme [16], superoxide-dismutase 1 [17] or S100A9 [18,19] have also been shown to be engaged in this type of cross-interaction. The wide variety of different proteins and peptides involved in this process suggests that the amyloid interactome is still far from fully understood. Recently, it has been observed that a cross-interaction may also exist between Tau and prion proteins [20,21,22], which requires a more in-depth examination and is the main subject of this work.

Intrinsically disordered microtubule associated protein Tau has 6 isoforms being expressed mainly in neurons with varying levels during the human lifetime [23,24]. Besides serving as microtubule dynamics regulators, Tau proteins also have many other important physiological functions, such as DNA stabilization, regulating neuronal polarity and axonal transport as well as participating in signaling pathways through the N repeat region [4,25]. However, when Tau protein is hyperphosphorylated or cleaved by several proteases, it starts to aggregate into Tau fibrils inside the neuron causing microtubule collapse, neuronal transport impairment and finally neuronal death [26]. After the fibrils reach the extracellular environment, they form neurofibrillary tangles which are one of the two main hallmarks of Alzheimer’s disease [6]. Although in this case the disease initiator is known to be amyloid-β plaques, presence of Tau tangles in various parts of the brain correspond to Alzheimer’s progression better [27].

Prion proteins are cell-surface glycoproteins, considered to play a role in several physiological functions, including copper homeostasis, neuroprotection and stem cell renewal [28]. Their aggregation into fibrillar aggregates is associated with multiple neurodegenerative disorders, such as Creutzfeldt–Jakob disease, Gerstmann–Straussler–Scheinker syndrome or fatal familial insomnia [29,30]. Prion proteins are well known for their ability to aggregate into multiple different type fibrils [31], with specific secondary structures [32], self-replication tendencies [33] and infectivities [34]. This property provides an opportunity to examine possible protein cross-interaction in vitro using multiple distinct prion protein fibril conformations.

In this work, we generated five conformationally distinct prion protein fibrils and examined their interaction with Tau proteins. We observed that there was a conformation-specific association between Tau monomers and prion protein fibrils, which increased their self-association and amyloidophilic dye binding capacity. We also quantified the Tau protein interaction with the five distinct prion protein fibril types and determined that this type of association did not cause the formation of Tau amyloid aggregates.

## 2. Results

To examine the possible interaction between Tau proteins and different prion protein (PrP) fibril conformations, it was first necessary to obtain multiple distinct PrP aggregate types. In order to achieve this, several samples of monomeric PrP were aggregated under two temperature conditions (25 °C and 60 °C), which are known to result in a random distribution of several structurally-unique fibril types [32]. The generated aggregates were examined by Fourier-transform infrared spectroscopy (FTIR), grouped based on similarities in their FTIR spectra and subjected to two rounds of self-replication to obtain a higher quantity of fibrils. The replicated PrP aggregates were examined by FTIR again, in order to verify that they maintained their initial structural differences (Figure 1). Based on the Amide I/I’ region FTIR second derivative minima positions (Figure 1 table insert), there were five PrP aggregate groups showing distinct beta-sheet hydrogen bonding and turn/loop motif profiles [35]. There was also a 4–6 cm^−1^ difference between the half-height bandwidth [36] of type 1–3 and type 4, 5 fibrils (Figure 1 table insert). These results displayed that all five of the selected PrP aggregates had differences in their secondary structures, which were retained after two rounds of reseeding.

To determine if there were also morphological distinctions between the five samples, they were additionally examined using atomic force microscopy (AFM). It was observed that, similarly to the previously shown aggregates generated under these conditions [32], they were highly fragmented and had a tendency to form large clusters (Figure A1). Out of the five selected samples, type 1 fibrils were least prone towards self-association and were more evenly spread out (Figure A1A), while all other four types formed 1–2 µm size clusters, composed out of short fibril fragments (Figure A1B–E). Due to this factor, it was not possible to gain additional information regarding the formed fibril morphological characteristics.

When all five PrP fibril types were combined with an equimolar concentration of Tau protein (25 µM), their association was tracked by measuring the change in fluorescence intensity of an amyloid specific dye-thioflavin-T (ThT) [37]. Due to self-association, settling and dye binding, the control samples also experienced a gradual increase in fluorescence intensity (Appendix B, Figure A2). To account for this factor, the control sample intensities were subtracted from the PrP-Tau sample fluorescence values. Tau protein, by itself, did not cause any changes to the ThT signal intensity (Appendix B, Figure A2F). In the case of type 1, 2, 4 and 5 fibrils, an increase in fluorescence intensity was observed during the first ~20 min, after which it reached a plateau (Figure 2A,B,D,E). The type 3 fibril interaction was the most peculiar, as the initial PrP-Tau fluorescence intensity was below the control sample and then eventually settled at a value close to the control (Figure 2C). Fibril type 4 and 5 samples had an initial fluorescence intensity difference even at the start of the measurement (Figure 2D,E), which suggested that their association occurred rapidly during the preparation procedure, even before the first plate scan.

Comparing the relative ThT fluorescence intensities of control and PrP-Tau samples at the end of the reaction (Figure 2F), it appeared that only fibril type 3 did not significantly deviate from the control, after it was combined with Tau protein (ANOVA Bonferroni means comparison, n = 8, *p* < 0.01). A statistical analysis of all other cases revealed that only the type 2 and type 4 fibril pair had significantly different relative signal intensities (n = 8, *p* < 0.01).

Despite a brief but vigorous sample agitation prior to each measurement, a visual inspection revealed that samples, which contained Tau, were considerably more opaque and had larger aggregate clusters than their controls. This suggested that the observed ThT fluorescence intensity changes could be caused by either the formation of Tau protein aggregates or their association with PrP fibrils, resulting in the formation of larger structures, which settled at the bottom of the 96-well plate. To rule out the latter possibility, the samples were homogenized by repetitive pipetting for 10 s after which their fluorescence and absorbance spectra were immediately acquired.

Interestingly, all five fibril types experienced a substantial increase in their optical density values at 600 nm (OD_600_) (Figure 2G) when they were combined with Tau proteins. Despite different absolute values, the relative change in OD_600_ was between 2.2 and 2.5 for all cases, with type 2 having significantly different values from the other four types (n = 3, *p* < 0.01). The fluorescence intensity value differences of samples with and without Tau protein (Appendix B, Figure A3) retained similar tendencies as were shown previously (Figure 2F), apart from a less substantial distinction for type 2 fibrils. Since both the initial, as well as agitated samples displayed an increase in ThT fluorescence intensity, this suggested that the structures formed in the presence of Tau protein could bind a higher number of dye molecules. The concentration of bound ThT was determined by pelleting the samples and scanning their supernatant ThT-specific absorbance values. In all cases, the concentration of bound dye molecules increased (Figure 2H), with the highest additional binding observed for type 4 and 5 PrP aggregates. Interestingly, while type 3 fibrils did not display any enhanced fluorescence upon the addition of Tau monomers, the PrP-Tau pair had more bound ThT molecules than its control counterpart.

The substantial increase in sample optical density, as well as ThT fluorescence intensity and bound dye molecules all suggested either the formation of Tau amyloid fibrils or an association between PrP fibrils and Tau monomers. In order to examine the structural changes that occurred during this event, FTIR spectra of all five PrP-Tau samples were acquired and compared to their respective control spectra. In all cases (Figure 3A–E), there was a notable increase around ~1650 cm^−1^ (Appendix B, Figure A4, associated with random coils [34]), with varying degrees among different fibril types. Such a change was to be expected, as both aggregated and non-aggregated Tau protein possess large, disordered sections [37].

Surprisingly, all five PrP-Tau FTIR spectra could be decomposed into monomeric Tau and the initial PrP fibril spectra, with no additional beta-sheet content or other secondary structure changes detected (Figure 3). The decomposition also revealed that the Tau protein part of the PrP-Tau spectra ranged from 16% to 12%, based on the type of initial PrP fibril (Figure 3F). This indicated that, on average, one Tau protein associated with 14–20 PrP monomers in their aggregated form. Taking into account such a relatively small number of bound Tau monomers and the apparent lack of any change in their secondary structure, these results suggested that the PrP-Tau interaction did not cause the formation of Tau amyloid aggregates. Considering that significant optical density changes of PrP fibril samples were previously observed to be caused by alterations in solution ionic strength [38], such PrP-Tau associations could also be linked to electrostatic effects.

To test this hypothesis, the PrP-Tau association reactions were repeated under four different ionic strength solutions, using fresh batches of type 1 and 5 PrP fibrils. The resulting OD_600_ (Figure 4A,B), fluorescence intensity (Figure 4C,D) and percentage of bound Tau monomers (Figure 4 table insert) all followed a similar downward trend with increasing NaCl concentration. The trend was also similar for both types of PrP fibrils, indicating a similar effect caused by a change in solution’s ionic strength. This relationship between PrP-Tau association and the concentration of sodium and chloride ions suggested that the interaction was most likely of electrostatic nature and Tau monomers adsorbed to the surface of prion protein fibrils.

## 3. Discussion

Based on all the data acquired in this work, it is quite clear that there is an interaction between prion protein fibrils and Tau protein monomers. The type of interaction, however, is quite peculiar and requires multiple assays to be determined. If we take into consideration the increase in ThT fluorescence intensity, the shift in sample optical density, higher concentration of bound ThT molecules and the association kinetics, it would imply the formation of additional amyloid fibrils. Despite this, the decomposition of FTIR spectra revealed that all five PrP-Tau samples were composed entirely of the initial PrP fibrils and Tau monomers, with no additional beta-sheets or secondary structure motifs present.

A possible explanation for these conflicting observations is a non-amyloid association between PrP fibrils and Tau monomers. Based on the ionic strength assay, the level of this interaction was reduced by the presence of sodium and chloride ions in solution, which implies the association may be of electrostatic nature. Such an interaction has also recently been reported for PrP and Tau protein in the formation of multiphasic condensates [22], which further supports the possible electrostatic nature of this association. The Tau 2N4R isoform used in this work has a theoretical pI of 8.24, while for monomeric MoPrP89-230 the pI is 9.06, which would imply a similar charge under the reaction conditions (pH 7.4). However, since the prion protein is in its aggregated state, this would result in a different distribution of amino acids on the surface of the fibril and change the overall surface charge of the structure. If Tau monomers mediate the interaction between PrP fibrils and cause larger aggregate formations, this would explain the increase in sample optical density. Such clumping events would also contribute to the shielding/entrapping of additional ThT molecules and result in a higher fluorescence intensity value [39]. The Tau monomers would then also be present in these larger aggregates, without actually altering their own secondary structure or forming additional beta-sheets, as was seen during the FTIR assay.

This type of association may have significant implications both during in vitro examinations, as well as in vivo. If a study only uses ThT-binding parameters, changes in optical density and measurements of non-aggregated protein concentrations, it will come to a false-positive conclusion of Tau protein amyloid aggregation on PrP fibrils, which is clearly not the case. In vivo, such interactions could be one of several possible explanations for the formation of large prion protein aggregate plaques, especially since both proteins share a localization and a relatively small concentration of Tau is needed to cause PrP fibril self-association. This may also apply to other amyloidogenic proteins, especially since one of the most well documented co-aggregation is between the Alzheimer’s disease-related amyloid-beta peptide and Tau protein [10].

Another interesting aspect is that the PrP and Tau association appears to be related to the conformation of the PrP fibrils. While certain aggregate types experienced an increase in bound-ThT fluorescence intensity (Figure 2, type 5), others had no observable changes (Figure 2, type 3), despite binding a comparable number of Tau monomers. The distinct PrP fibril types were also capable of associating with a different number of Tau monomers, with type 1 aggregates binding the largest amount and type 5, the smallest. Coincidentally, the type 1 aggregates were also the least prone towards self-association, as was observed in the AFM images of all five samples (Appendix B, Figure A1A). This phenomenon may be related to the prion protein fibril surface charge distribution [40], which could be affected by their distinct secondary structures. It is known that variations in the beta-sheet hydrogen bonding (as can be seen in the sample FTIR spectra) can influence amyloid aggregate morphologies and, in turn, their surface motifs [41,42,43]. Distinct surface charge distributions would explain their specific capacity for Tau monomers and why an increase in solution ionic strength mitigated this electrostatic interaction.

Taking everything into consideration, it seems that prion protein fibrils have conformation-specific interactions with Tau proteins, which increase their self-association tendencies, as well as amyloid-specific dye incorporation properties. This interaction appears to rely on electrostatic effects and does not trigger Tau protein amyloid aggregation on prion protein fibrils.

## 4. Materials and Methods

### 4.1. Prion Protein Aggregate Preparation

Mouse prion protein fragment 89–230 (further referred to as PrP) was purified as described previously [44] without the His-tag cleavage step. The protein was then exchanged into a 10 mM sodium acetate buffer solution (pH 4.0) by dialysis, concentrated to 4 mg/mL and stored at −20 °C. Prior to aggregation, PrP was thawed at room temperature and combined with 10× PBS, 1× PBS with 6 M guanidine hydrochloride (GuHCl, pH adjusted to 7.4) and MilliQ H_2_O to a final protein concentration of 25 µM, 1× PBS and 2 M GuHCl. The reaction solution was then distributed to 96-well half-area non-binding plates (cat. No 3881, Corning Incorporated, Corning, NY, USA) (6 repeats each, 100 µL final volume) and 3 mm glass beads were added to each well. The plates were then incubated at either 25 °C or 60 °C in a Clariostar Plus plate reader with constant 600 RPM agitation (72 h at 25 °C or 24 h at 60 °C). The generated aggregate solutions (100 µL) were then combined with the initial reaction solution (400 µL) and incubated under their respective conditions (600 RPM agitation, 72 h at 25 °C or 24 h at 60 °C) in 2.0 mL test-tubes (each containing two 3 mm glass beads). Afterwards, an additional 1.5 mL of initial reaction solution were added to each test-tube and they were incubated for the same amount of time.

Then, 1 mL of each aggregate solution was examined using FTIR (as described in the Fourier-transform infrared spectroscopy section) and the samples were grouped based on differences in their spectra. Five different fibril type samples (1 mL) were combined with 4 mL initial reaction solution and incubated under their respective conditions. The procedure was repeated again with the addition of 10 mL initial reaction solution, to yield a final 15 mL volume of each aggregate type.

Next, 1 mL of each aggregate type solution was examined using FTIR, while the remaining 14 mL were centrifuged at 12,500 RPM for 10 min and resuspended into 50 mM Hepes (pH 7.4) buffer solution. This centrifugation and resuspension procedure was repeated 3 times. The concentrations of non-aggregated prion proteins were determined by scanning the absorbance of the first supernatants at 280 nm (ε_280_ = 27,515 M^−1^cm^−1^). During the final resuspension step, the aggregate pellets were resuspended into specific volumes of Hepes buffer solutions, which would result in all samples having identical 50 µM concentrations of proteins in their aggregated state (after taking into account the non-aggregated protein concentrations).

### 4.2. SDS-PAGE

To examine if the final fibril solutions contained any residual monomeric or oligomeric components, the aggregate samples (30 µL) were combined with a 4 times concentrated SDS-PAGE sample buffer solution (10 µL) and incubated at room temperature for 15 min. Heating was not applied to avoid possible fibril destabilization by SDS. The samples were then loaded on a 12% acrylamide gel (7 µL of each sample, 5 µL Pierce unstained protein marker (cat. No 26610, Fisher Scientific USA, Hampton, NH, USA). The gel was stained using GelCode Blue Safe Protein Stain (cat. No 10763505, Fisher Scientific USA, Hampton, NH, USA). The resulting gel did not contain any notable monomeric or oligomeric prion protein forms (Appendix B, Figure A5).

### 4.3. Tau Protein Purification

Recombinant Tau protein 2N4R isoform was purified as described previously [45] with a difference in one protein expression step. Briefly, pET302/NT-His-SUMO-Tau plazmids were transformed in *E. coli* BL21(DE3) OneStar cells and cultured on a Petri plate with LB agar supplemented with 50 mg/mL kanamycin and incubated overnight at 37 °C. A colony was inoculated into 50 mL LB supplemented with 50 mg/mL kanamycin and grown overnight at 37 °C. An overnight culture was used for 100-fold inoculation of 3.6 L Terrific Broth (TRB) medium [46] supplemented with 50 mg/mL kanamycin and 1% ethanol [47] which then was incubated at 37 °C until A_600_ 2.0 and then expression was induced with 2 mM IPTG. The resulting medium was incubated at 37 °C for 5 h. Cells were harvested by centrifugation at 6000× *g*, 4 °C for 30 min. All further purification procedures were completed as described previously [45]. After gelfiltration, Tau protein was exchanged to 50 mM Hepes, pH 7.4 buffer using a HiTrap Desalt column, supplemented with 20 mM NaCl, concentrated to 4 mg/mL and stored at −80 °C.

### 4.4. Prion Protein Fibril and Tau Monomer Association

PrP fibril and Tau monomer solutions were combined with a 10 mM thioflavin-T (ThT) stock solution to yield 25 µM PrP fibrils, 25 µM Tau monomer and 100 µM ThT. Control samples contained an equivalent volume of 50 mM Hepes buffer (pH 7.4) solution, containing 20 mM NaCl, in place of the Tau protein solution. The reaction solutions were then immediatelly placed into 96-well plates (8 repeats each, 100 µL final volume) and incubated at 25 °C. The fluorescence intensity of ThT (440 nm excitation and 480 nm emission wavelengths) was measured every 2 min with 10 s of 600 RPM orbital agitation prior to each measurement cycle. The fluorescence intensity difference was determined by subtracting the control sample intensity values from the PrP-Tau solution intensities. For reactions under different ionic strength conditions, the reaction solutions contained an additional 50 mM, 100 mM or 200 mM NaCl, which was added by replacing a part of the reaction buffer with a 50 mM Hepes buffer (pH 7.4) solution, containing 2 M NaCl. After the association reactions, 8 repeats of each condition were combined to a final volume of 800 µL. The combined solutions were then used for further analysis.

### 4.5. Fluorescence and Absorbance Measurements

Before fluorescence and absorbance measurements, each sample was homogenized by pipetting for 10 s. ThT fluorescence intensity was measured using a Varian (Agilent, Santa Clara, CA, USA) Cary Eclipse fluorescence spectrometer, with 440 nm excitation and 480 nm emission wavelengths (5 nm excitation, 2.5 nm emission slit widths) in a 3 mm pathlength cuvette. Absorbance measurements were conducted using a Shimadzu (Kyoto, Japan) UV-1800 spectrophotometer in the range from 200 nm to 600 nm (1 nm steps) in a 3 mm pathlength cuvette. Three measurements were taken for each condition (sample volume was 100 µL). The values obtained at 600 nm were regarded as sample optical densities (OD_600_).

To determine the concentration of free ThT molecules under each condition, the samples were centrifuged at 12,500 RPM for 10 min, after which 100 µL of each supernatant was carefully removed. The concentration of residual ThT in each sample was calculated by scanning the supernatant absorbance at 412 nm (ε_412_ = 23,250 M^−1^cm^−1^). Based on the obtained free ThT concentration values, the difference between bound dye concentrations of PrP and PrP-Tau samples was determined.

### 4.6. Fourier-Transform Infrared Spectroscopy

500 µL of each sample was centrifuged at 12,500 RPM for 10 min, after which the supernatant was removed and replaced with 200 µL of D_2_O, containing 400 mM NaCl (addition of NaCl improved prion protein fibril sedimentation [38]). The centrifugation and resuspension procedures were repeated 4 times. During the final resuspension, the aggregate pellet was mixed with 30 µL of D_2_O with NaCl. The suspension was then placed between two CaF_2_ transmission windows, with a 0.05 mm Teflon spacer and 256 interferograms were scanned using a Bruker (Billerica, MA, USA) Invenio S FTIR spectrometer. D_2_O and water vapor spectra were then subtracted from the sample spectra, which were then baseline corrected and normalized between 1595 cm^−1^ and 1700 cm^−1^. All data processing was conducted using GRAMS software.

To acquire a Tau monomer spectrum, the protein solution was exchanged into a D_2_O solution (containing 50 mM NaCl) by concentrating the sample using a 10 kDa concentrator, diluting it with D_2_O and repeating this concentration and dilution procedure 4 times. The composition of PrP-Tau sample spectra was determined by combining separate PrP fibril and Tau monomer spectra and comparing it to the PrP-Tau spectra, using the least squares method.

### 4.7. Atomic Force Microscopy

For each aggregate solution, 0.5 mL was briefly sonicated for 10 s using a Bandelin Sonopuls ultrasonic homogenizer, equipped with a MS-72 sonication tip (constant sonication, 20% of total power). Then, 30 µL aliqouts of each solution were placed on freshly cleaved mica and incubated for 2 min at room temperature. The mica were then washed with 2 mL of MilliQ H_2_O and dried using airflow. AFM images of the aggregates were acquired using a Dimension Icon atomic force microscope (Bruker, Billerica, MA, USA) as described previously [48]. AFM images were analyzed using Gwyddion 2.57 software.

## Figures and Tables

**Figure 1 ijms-24-09277-f001:**
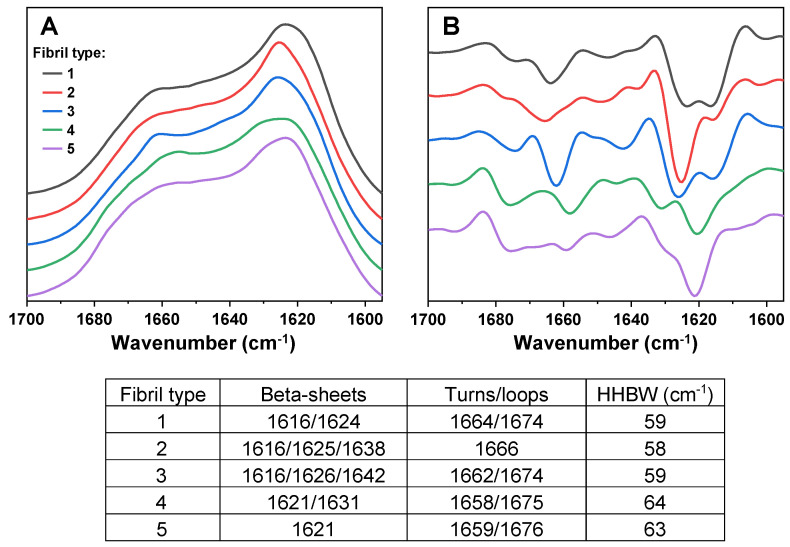
FTIR spectra (**A**) and their second derivatives (**B**) of PrP fibrils, generated under 25 °C (fibril types 1 and 2) and 60 °C (fibril types 3, 4 and 5) temperature conditions. FTIR spectra acquisition is described in the Section 4. The table insert displays the second derivative minima positions and the FTIR spectra HHBW values. FTIR spectra raw data are available as Appendix A.

**Figure 2 ijms-24-09277-f002:**
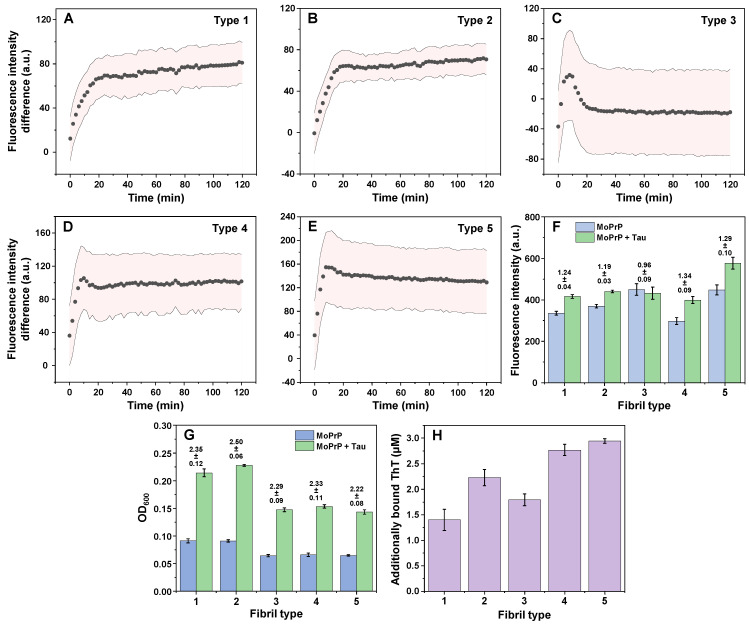
Prion protein fibril association with Tau monomer kinetics and resulting sample optical properties. ThT fluorescence intensity differences between control and PrP-Tau samples over the course of 120 min for type 1 (**A**), 2 (**B**), 3 (**C**), 4 (**D**) and 5 (**E**) PrP fibrils. In each case, the light red plot indicates the combined error of the control and PrP-Tau samples (one standard deviation, 8 repeats, data points are the average value of 8 repeats). Fluorescence intensity absolute values (**F**) and their relative differences (indicated above bar graphs, n = 3, error bars are for one standard deviation). Sample optical densities after agitation (**G**) and concentrations of additionally bound ThT molecules ((**H**), n = 3, error bars are for one standard deviation). Sample preparation and fluorescence/absorbance measurement procedures are described in the Section 4. All raw data are available as Appendix A.

**Figure 3 ijms-24-09277-f003:**
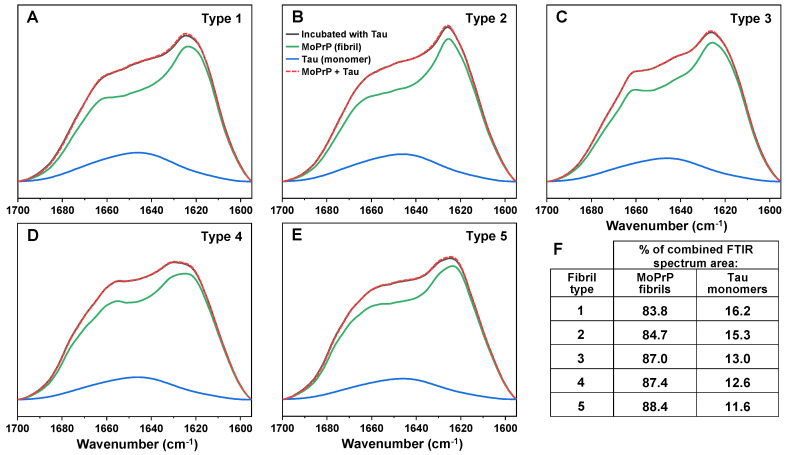
FTIR spectra of PrP-Tau samples (black line) and their decomposition into PrP fibril (green line) and Tau monomer (blue line) spectra ((**A**)−type 1, (**B**)−type 2, (**C**)−type 3, (**D**)−type 4, (**E**)−type 5). Comparison of decomposed spectra sum to original spectrum is shown as red dashed line. Table insert (**F**) displays the decomposition values of each PrP-Tau sample, based on the % of amide bonds, contributing to the total spectrum (PrP−163, Tau monomer−440 amide bonds). FTIR spectra acquisition and decomposition procedures are described in the Section 4.

**Figure 4 ijms-24-09277-f004:**
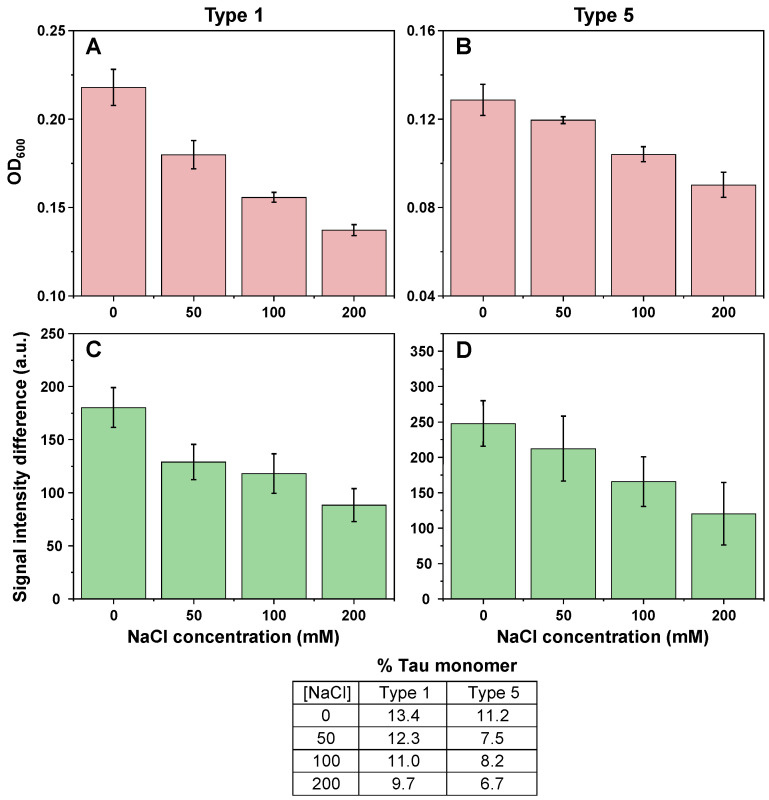
Optical density (**A**,**B**) and ThT fluorescence intensity differences (**C**,**D**) of type 1 and type 5 fibrils, when incubated with Tau monomers in the presence of additional 0, 50, 100 and 200 mM of NaCl (n = 3, error bars are for one standard deviation). Table insert displays the decomposition values of type 1 and 5 PrP-Tau samples, based on the % of amide bonds, contributing to the total FTIR spectrum (PrP−163, Tau monomer−440 amide bonds). Optical density, fluorescence intensity and FTIR spectra scanning procedures are described in the Section 4. All raw data are available as Appendix A.

## Data Availability

All data are available in the Appendix A.

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
