# Peer review of "Conformation-Specific Association of Prion Protein Amyloid Aggregates with Tau Protein Monomers"

_ijms, 2023, doi:10.3390/ijms24119277_

Round 1

Reviewer 1 Report

In this study, Smirnovas and coworkers provide information on the cross-interactions between Tau and prion amyloidogenic proteins, during the fibril formation process. In particular, they have generated five populations of conformationally distinct prion protein amyloid fibrils and examined their interaction with Tau proteins using a wide range of spectroscopic techniques. In this respect, they show that prion protein fibrils possess conformation-specific interactions with Tau proteins, which increase their self-association tendencies. In addition, this interaction appears to rely on electrostatic effects and does not trigger Tau protein amyloid aggregation on prion protein fibrils.

This study provide important insights that contribute to the understanding of the molecular basis of amyloid aggregation. The paper is well written, the reference section is exhaustive, and the goal of the research is clear. For the above reasons, in my opinion it is suitable for publication in “International Journal of Molecular Sciences”.

Reviewer 2 Report

 The study examines a possible interaction involving Tau and prion proteins. Five populations of distinct prion protein amyloid fibrils were prepared and incubated with Tau monomers. The apparent lack of change in the secondary structure suggests that the PrP-Tau interaction did not lead to the formation of Tau or PrP amyloid aggregates but rather suggests a non-amyloid electrostatic association between PrP fibrils and Tau monomers. On the opposite, it supports the idea that fibrils are  not involved and that tau and prion proteins can commingle into multicomponent-like liquid-liquid condensates (S.K. Rai et al.PNAS-2023)

 Specific comments.

Figure 1-IR spectra do not prove that we are dealing with fibrils but rather with a mixture of monomers, oligomers, and fibrils. Several IR spectra reveal indeed the formation of oligomers (1685cm-1) suggesting that we are still in the process of fibrils formation. Also changes in Thioflavine-T fluorescence illustrate that fibrils have been formed but do not rule out the presence of monomers or oligomers. Identification of all species (monomer, oligomer, fibril) by Western blot analysis on a 12% Bis-Tris SDS/PAGE gel followed by antibody recognition is a prerequisite for any rational interpretation of the data

 Figure 3A-E. The authors mention a notable increase at  ~1650 cm-1 (associated with random coils). How was such an increase measured? Can one rule out that such an increase around 1650 cm-1 is representative of helical domains. H/D exchange measurements will address the question elegantly. The apparent lack of change in the secondary structure suggested that the PrP-Tau interaction did not cause the formation of Tau or PrP amyloid aggregates.

Reviewer 3 Report

1-It is recommended to use another dye like Congo Red or Nile Red to detect the amyloid fibrils.

2-The authors did not provide any AFM or TEM images to distinguish among five types of prion amyloid fibrils.

3- The authors should provide more explanation about that how electrostatic interaction between tau monomers and prion protein results in increased fibrillation of prion protein.

4-It is recommended to perform molecular docking or molecular dynamics simulation to provide more molecular details about the involved interactions between tau monomers and prion proteins.

Reviewer 4 Report

In this manuscript by Ziaunys et al, the experiments are designed appropriately and conducted carefully. Also, the manuscript is written very well, and the interpretation of the results by the authors is sound.

One thing that is missing from this manuscript is statistical analysis. It is unclear how the authors evaluated statistical significance of their data. The authors mention several times that changes observed are within margin of error, but the basis for this kind of statement is not provided in the result or materials and methods.

Also, authors need to report the number of replicates for all the experiments shown in the manuscript. Please also clearly state what each bar represents (mean? median?) and what each error bar represents (sem? sd?) in corresponding figure legends.

Round 2

Reviewer 2 Report

.

Reviewer 3 Report

All of the required responses are provided.